# Cost-effectiveness of the ReDIRECT/ counterweight-plus weight management programme to alleviate symptoms of long COVID

Heather L. Fraser [1] ✉, Laura Haag [2], Naomi Brosnahan[2,3], Alex McConnachie [4], Janice Richardson[5], Caroline E. Haig[4], Tracy Ibbotson [6], Jane Ormerod[7], Catherine A. O'Donnell[6], Michael E. J. Lean [2], Naveed Sattar [5], David N. Blane [6], Emilie Combet[2] & Emma McIntosh[1]

Long-term effects of COVID-19 infection, termed Long COVID (LC), are associated with reduced quality of life. Symptoms associated with overweight/obesity overlap with and may aggravate those of LC. This paper reports the economic evaluation alongside the ReDIRECT Trial, which evaluated the impact of an evidence-based, remotely-delivered weight management programme on self-reported symptoms of LC in those living with overweight/obesity in the United Kingdom. Recruited participants (*n* = 234) were randomly allocated to the intervention group (weight management) or control group (usual care). Incremental costs and Quality-Adjusted Life Years (QALYs) were calculated using intervention cost, healthcare resource use and EQ-5D-5L data collected at baseline, three and 6 months. In this work, we show that the ReDIRECT intervention is likely cost-effective in improving LC symptoms from an NHS/PSS perspective, compared to usual care (Incremental Cost-Effectiveness Ratio of £14,754/QALY). Adopting a broader societal perspective, the intervention becomes potentially cost saving compared to usual care.

Long COVID (LC), described as persistent symptoms following COVID-19 infection, places a large burden on health systems globally[1], with those affected suffering a wide range of symptoms, including fatigue, breathlessness, cognitive dysfunction and widespread pain[2,3]. Although research into treatment of LC is ongoing, there are no established interventions, other than self-management, support and information, and treatment of individual symptoms[4,5]. A Bayesian meta-regression of 54 studies and two databases estimated that, of individuals surviving symptomatic episodes of COVID-19 infection, 6.2% experienced at least one LC symptom 3 months after initial infection, with 15.1% of these individuals experiencing persistent symptoms at 12 months[6]. Studies with matched controls conducted in the UK and the Netherlands suggest its prevalence is between 6 and 12% three to six months after initial COVID-19 infection, with UK estimates suggesting 10% prevalence of symptoms up to 18 months after infection[7,8].

[1]Health Economics and Health Technology Assessment, School of Health and Wellbeing, College of Medical, Veterinary and Life Sciences, University of Glasgow, Glasgow, UK. [2]Human Nutrition, School of Medicine, Dentistry & Nursing, College of Medical, Veterinary and Life Sciences, University of Glasgow, Glasgow, UK. [3]Counterweight Ltd, London, UK. [4]Robertson Centre for Biostatistics, School of Health and Wellbeing, College of Medical, Veterinary and Life Sciences, University of Glasgow, Glasgow, UK. [5]School of Cardiovascular and Metabolic Health, College of Medical, Veterinary and Life Sciences, University of Glasgow, Glasgow, UK. [6]General Practice and Primary Care, School of Health and Wellbeing, College of Medical, Veterinary and Life Sciences, University of Glasgow, Glasgow, UK. [7]Long Covid Scotland, Aberdeen, UK. ✉e-mail: heather.fraser@glasgow.ac.uk

**Table 1 | Population characteristics in the two groups in the randomised trial**

| | Control (N = 118) | Intervention (N = 116) | Overall (N = 234) |
|---|---|---|---|
| **Mean age (SD)** | 46.1 (10.5) | 46.4 (9.14) | 46.3 (9.85) |
| **Sex** | | | |
| Female | 101 (85.6%) | 99 (85.3%) | 200 (85.5%) |
| Male | 17 (14.4%) | 17 (14.7%) | 34 (14.5%) |
| **Mean weight in kg at baseline (SD)** | 102 (19.4) | 102 (21.5) | 102 (20.4) |
| **Mean body mass index at baseline (SD)** | 36.8 (6.49) | 36.8 (7.47) | 36.8 (6.98) |
| **Utility at baseline\*** | | | |
| Mean (SD) | 0.474 (0.272) | 0.482 (0.237) | 0.478 (0.255) |
| Median [min, max] | 0.532 [−0.246, 0.985] | 0.516 [−0.156, 0.985] | 0.530 [−0.246, 0.985] |
| **Primary symptom (participant-selected)** | | | |
| Pain | 14 (11.9%) | 14 (12.1%) | 28 (12.0%) |
| Breathlessness | 17 (14.4%) | 20 (17.2%) | 37 (15.8%) |
| Fatigue | 66 (55.9%) | 60 (51.7%) | 126 (53.8%) |
| Anxiety/depression | 2 (1.7%) | 1 (0.9%) | 3 (1.3%) |
| Other | 19 (16.1%) | 21 (18.1%) | 40 (17.1%) |
| **Ethnicity** | | | |
| South Asian | 5 (4.2%) | 5 (4.3%) | 10 (4.3%) |
| Other Asian or Asian British background | 0 (0%) | 5 (4.3%) | 5 (2.1%) |
| Black, African, Caribbean or Black British | 2 (1.7%) | 0 (0%) | 2 (0.9%) |
| White | 105 (89.0%) | 106 (91.4%) | 211 (90.2%) |
| Other or mixed ethnic group | 6 (5.1%) | 0 (0%) | 6 (2.6%) |
| **Index of multiple deprivation (quintile)** | | | |
| Quintile 1 (most deprived) | 13 (11.0%) | 18 (15.5%) | 31 (13.2%) |
| Quintile 2 | 22 (18.6%) | 22 (19.0%) | 44 (18.8%) |
| Quintile 3 | 29 (24.6%) | 16 (13.8%) | 45 (19.2%) |
| Quintile 4 | 28 (23.7%) | 23 (19.8%) | 51 (21.8%) |
| Quintile 5 (least deprived) | 26 (22.0%) | 37 (31.9%) | 63 (26.9%) |
| **Employment status at baseline** | | | |
| Full-time employment | 50 (42.4%) | 51 (44.0%) | 101 (43.2%) |
| Part-time employment | 35 (29.7%) | 34 (29.3%) | 69 (29.5%) |
| Retired | 4 (3.4%) | 2 (1.7%) | 6 (2.6%) |
| Student | 0 (0%) | 1 (0.9%) | 1 (0.4%) |
| Unemployed | 7 (5.9%) | 10 (8.6%) | 17 (7.3%) |
| Other | 22 (18.6%) | 18 (15.5%) | 40 (17.1%) |
| **Mean healthcare resource use cost at baseline (SD)** | £652 (650) | £893 (1540) | £773 (1190) |
| **Productivity loss at baseline** | | | |
| Mean hours missed per week (SD) | 9.18 (13.3) | 11.7 (15.2) | 10.5 (14.3) |
| Mean productivity cost per week (SD) | £139 (200) | £170 (226) | £154 (214) |

\*Measured using the EQ-5D-5L, mapped to EQ-5D-3L utility values for the UK population, values range from −0.532 (extreme problems for all dimensions) to 1 (perfect health).

In addition to the clinical and quality of life burden, LC carries a substantial economic burden at both individual and societal levels[9–12]. LC has been found to affect labour participation, employment, and productivity of individuals as well as their caregivers[13]. Even in those not hospitalised with COVID-19, up to 23% of people remain absent from work between three and seven months after acute infection[11]. In addition to full absence from work, reduced capacity to work is also a factor[11], with the mean monthly income from work declining by 24.5% in a survey of people living with LC in the UK[14]. Therefore, evaluating interventions that may reduce the clinical and economic burden of LC are of societal importance[9,12,13].

Previous research has hypothesised that symptoms associated with overweight/obesity overlap with, and may aggravate, those of LC[15]. It is established that weight loss can improve health-related quality of life (HRQoL)[16–18] and may also lower levels of systemic inflammation[19–22], one potential factor in LC pathophysiology[23,24]. The Remote Diet Intervention to Reduce Long COVID Symptoms Trial (ReDIRECT) is a randomised controlled trial (RCT), which evaluated whether a remotely delivered weight loss intervention can help to alleviate the symptoms of LC in those living with overweight/obesity[2]. The intervention, Counterweight-Plus, is a dietary weight management programme, which the DiRECT trial showed to be safe, effective, and cost-effective in achieving and sustaining weight loss and remission of Type 2 Diabetes[25,26]. In the ReDIRECT RCT, the weight management programme was delivered remotely, using digital technology and remote dietetic advice[2].

This economic evaluation estimated the likely cost-effectiveness of the remotely delivered Counterweight-Plus intervention to alleviate LC symptoms. It is vital to understand the value for money gained from the intervention to inform decision-making on its implementation[27].

## Results

### Baseline characteristics
The characteristics of the study population at baseline are presented in Table 1, with further details reported elsewhere[15]. Participant characteristics, including mean age, weight, and utility were similar across the intervention and control groups.

### Health-related quality of life (HRQoL)
Utility scores for each participant were calculated using the EuroQol-5 Dimensions-5 Levels questionnaire (EQ-5D-5L) data, mapped to EQ-5D-3L utilities using the UK value set, and are presented in Fig. 1 by trial arm over the trial period. Figure 1a reports mean utility scores with standard error shading, while Fig. 1b shows the distribution of the utility scores in each arm, including the presence of outliers at the lower end of the utility score spectrum. Table 2 presents the summary results of the Area Under the Curve (AUC) utility score analysis (unadjusted and adjusted for baseline covariates), with total Quality-Adjusted Life Years (QALYs) calculated.

### Intervention micro-costing
Counterweight Ltd charges the NHS £1,000 per person for delivery of the intervention, which was therefore deemed the appropriate intervention cost to use in the base case analysis (rather than the micro-cost estimate). Micro-costing of the intervention, with cost components identified, measured, and valued resulted in a mean cost per participant of approximately £885 (base year 2022) — see Table S2 in the Supplementary Materials.

### Healthcare resource use
Figure 2a presents a line graph showing the mean cost per participant by trial arm at baseline, 3 months and 6 months, while Fig. 2b presents box plots of the distribution of this resource use over the same period. These plots illustrate the difference between the mean resource use at baseline in the two trial arms,

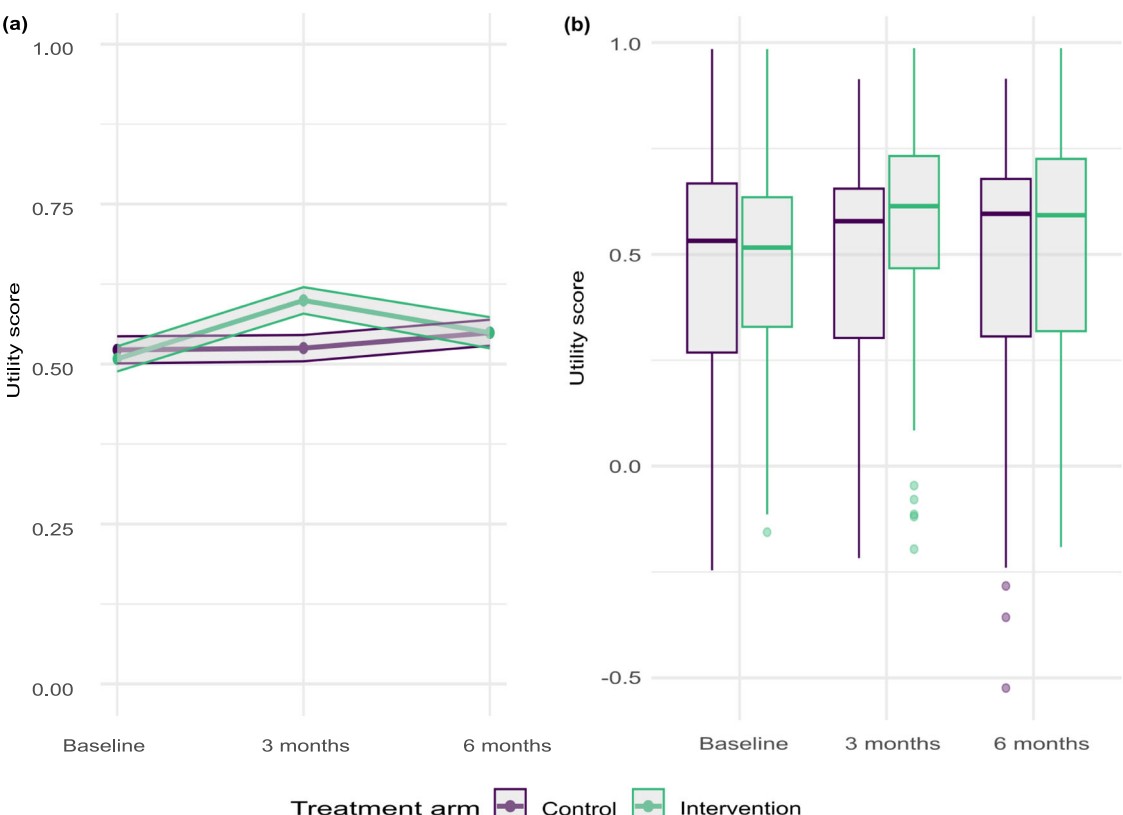

**Fig. 1 | Utility score from EQ5D questionnaire by trial arm at baseline, 3 months, and 6 months after randomisation. a** Line graph showing mean utility score by trial arm, with standard error shading (*n* = 118 control, *n* = 116 intervention), and **b** box plot showing the distribution of utility scores by trial arm over the trial period ($n = 118$ control, $n = 116$ intervention). The box plots show the median (centre line), the first and third quartiles (the lower and upper bounds of the box), and the whiskers show 1.5× the interquartile range. Points beyond the whiskers are 'outlying' points plotted individually. Source data are provided with this paper.

### Table 2 | AUC utility values and QALY results

| | Control | | Intervention | |
|---|---|---|---|---|
| | **Unadjusted** | **Adjusted*** | **Unadjusted** | **Adjusted*** |
| Utility at baseline | 0.474 | 0.469 | 0.482 | 0.478 |
| Utility at 3 months | 0.480 | 0.451 | 0.569 | 0.537 |
| Utility at 6 months | 0.484 | 0.458 | 0.527 | 0.500 |
| Total QALYs | 0.479 | 0.457 | 0.537 | 0.513 |

*Adjusted for: primary outcome selected, sex, age, index of multiple deprivation, ethnicity and primary outcome at baseline, in line with the statistical analysis conducted for the primary trial outcome.

as well as the presence of healthcare resource use outliers – typically resulting from hospitalisation in the 3 months preceding each assessment. One observation, found to have particularly high healthcare resource use at baseline (in the control arm), was subject to a further sensitivity analysis (outlier removed). This sensitivity analysis showed that excluding the outlier observation would not alter the interpretation of cost-utility analysis results (Supplementary Materials Tables S3 and S4 and Figs. S1 and S2). With no further indication to remove the outlier, this observation was included in the base case analysis. Differences in healthcare resource use in the intervention arm, relative to the control arm, were not statistically significant at 6 months (mean difference: -£47.03, 95% CI: -£236.83 to £142.77).

The healthcare resource use cost drivers are displayed in Fig. 2c, showing high hospitalisation costs at baseline and 3 months after randomisation for the control arm, relative to the intervention arm. Summary cost results (unadjusted and adjusted for baseline covariates) are presented in Table 3.

### Societal perspective: productivity losses

Figure 3a shows the total number of hours of work missed due to sickness in the week prior to each assessment, by trial arm, while Fig. 3b shows the mean number of hours of work missed due to sickness in each arm over the trial period. With weekly wage rates applied to hours missed due to sickness and extrapolated over the trial period, the costs associated with lost productivity are presented in Table 3, along with per-person food costs. The mean number of hours of work missed due to sickness reduced by 2.3 (SD 14.02) in the control arm and by 2.8 (SD 14.50) in the intervention arm, on average, with a treatment effect of 1.91 (95% CI −1.01 − 4.83, *p* = 0.20), adjusting for primary outcome selected, sex, age, index of multiple deprivation, ethnicity and hours of work missed due to sickness at baseline.

### Handling missing data

Data were assessed to be missing at random (MAR), with an imbalance in missingness by trial arm: 1 observation (out of 354) was missing in the control arm, while 36 observations (out of 348) were missing in the

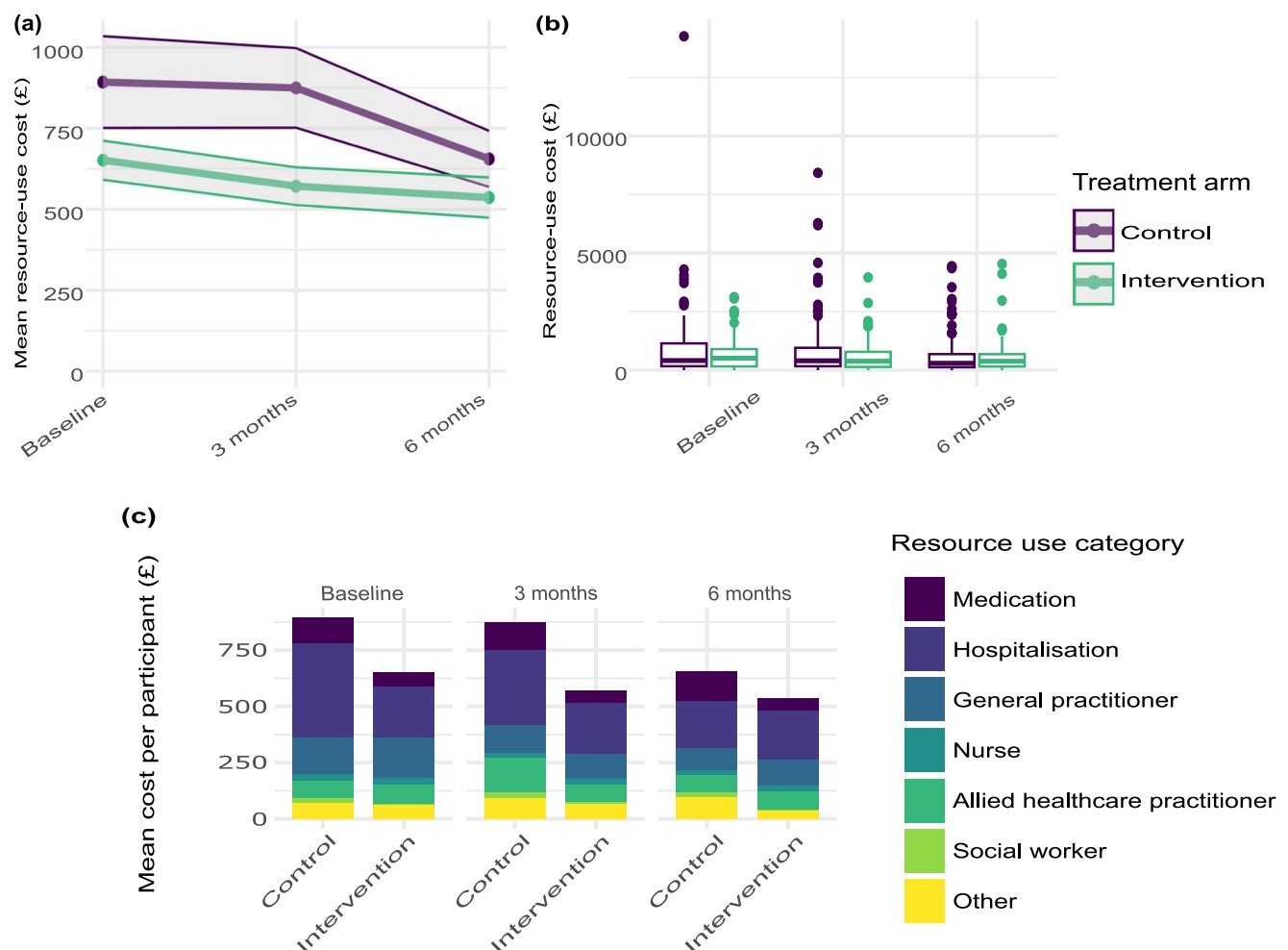

**Fig. 2 | Healthcare resource utilisation by trial arm at baseline, 3 months, and 6 months after randomisation. a** shows the mean resource use per participant over the trial period, by trial arm, with standard error shading (*n* = 118 control, *n* = 116 intervention); **b** shows box plots of the distribution of resource use costs among participants over the trial period, by trial arm (*n* = 118 control, *n* = 116 intervention). The box plots show the median (centre line), the first and third quartiles (the lower and upper bounds of the box), and the whiskers show 1.5× the interquartile range. Points beyond the whiskers are 'outlying' points plotted individually. **c** is a bar chart showing resource use category cost drivers over the trial period, by trial arm (*n* = 118 control, *n* = 116 intervention). Source data are provided with this paper.

**Table 3 | Summary cost results (£)**

| Perspective | Cost | Control | | Intervention | |
|---|---|---|---|---|---|
| | | Unadjusted | Adjusted* | Unadjusted | Adjusted* |
| NHS/PSS† | Intervention costs | 0 | 0 | 1000 | 1000 |
| | Healthcare resource use | 2423 | 2744 | 1759 | 2242 |
| Societal perspective | Productivity losses: trial duration | 5439 | 7778 | 4088 | 6715 |
| | Productivity losses: 7 days before assessment | 453 | 648 | 341 | 560 |
| | Food costs: trial duration | 2050 | 2853 | 1850 | 2646 |
| | Food costs: 7 days before assessment | 171 | 238 | 154 | 221 |
| NHS/PSS† | TOTAL | 2423 | 2744 | 2759 | 3242 |
| Societal | | 9912 | 12,223 | 8690 | 11,678 |

*Adjusted for: primary outcome selected, sex, age, index of multiple deprivation, ethnicity and all resource use at baseline, in line with the statistical analysis conducted for the primary trial outcome.
†National Health Service and Personal Social Services.

intervention arm, with a total of 25 individuals having at least one missing observation (1 out of 118 in the control arm and 24 out of 116 in the intervention arm). Multiple imputations using chained equations (MICE) was used to impute missing data with 10 iterations using the

*mice* function in R[28]. Seemingly unrelated regression (SUR) models controlled for age, sex, index of multiple deprivation, the participant-selected primary outcome, as well as costs and utility at baseline. Pooled cost difference and effect difference were estimated from

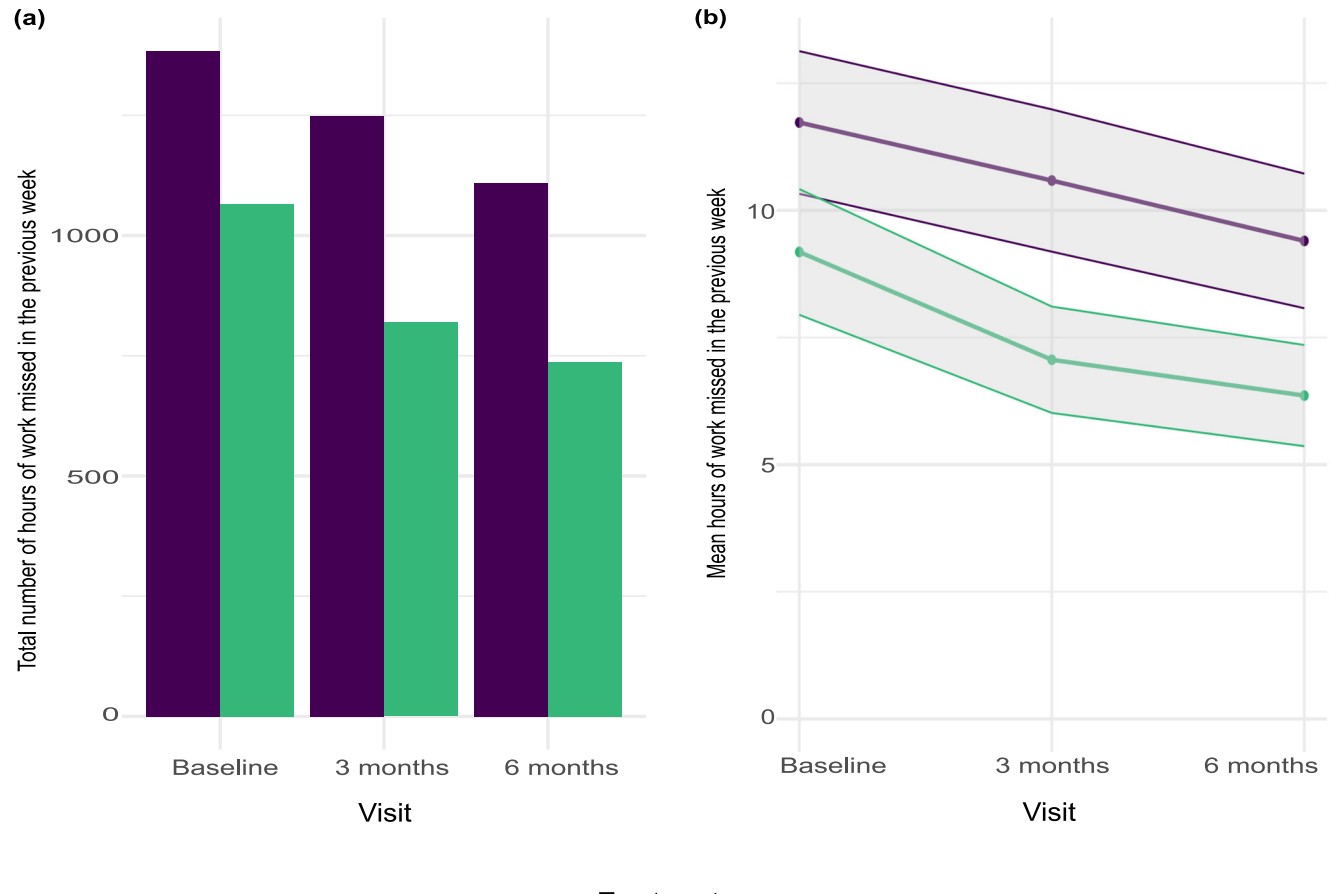

**Fig. 3 | Hours of work missed due to sickness by trial arm, at baseline, 3 months, and 6 months after randomisation (n = 118 control, n = 116 intervention). a** Bar chart showing the total number of hours of work missed, by trial arm; **b** line graph showing mean number of hours of work missed per participant, by trial arm, with standard error shading. Source data are provided with this paper.

5000 bootstrapped replicates of the imputed dataset using Rubin's Rules to estimate the incremental cost-effectiveness ratio (ICER) and 95% confidence intervals.

## Cost-effectiveness analysis

Figures 4a, c present ICER planes of the bootstrapped results from the National Health Service and Personal Social Services (NHS/PSS) perspective and societal perspective, respectively, while Fig. 4b, d present cost-effectiveness acceptability curves (CEACs) from the NHS/PSS and societal perspectives, respectively. In Fig. 4a, c, the diagonal lines crossing through the planes represent the £30,000/QALY cost-effectiveness threshold indicated for use by the National Institute for Health and Care Excellence (NICE)[29], with the corresponding CEACs - Fig. 4b, d - showing the probability of weight management being considered cost-effective at different cost-effectiveness thresholds. Figure 4b shows that at a threshold of £20,000/QALY, there would be approximately 72% probability of the intervention being considered cost-effective, increasing to 89% at a threshold of £30,000/QALY. From a broader societal perspective, Fig. 4d shows that there would be a 93% probability of the intervention being considered cost-effective at a threshold of £20,000/QALY, increasing to 96% at a threshold of £30,000/QALY. Further, Fig. 4c, d indicate that the intervention may be cost-saving when a societal perspective is adopted, with a 74% probability of being cost-saving at a threshold of £0/QALY.

Table 4 presents the pooled estimates, along with the 95% confidence intervals, for the between-group QALY and cost differences, from the NHS/PSS perspective and the broader societal perspective.

## Discussion

This paper reports the within-trial cost-utility analysis of the ReDIRECT trial, a remotely-delivered weight management programme to alleviate symptoms of LC[5]. In the base case analysis, taking an NHS/PSS perspective, we found that weight management is likely to be cost-effective, with an ICER of approximately £14,800/QALY and an 89% probability of cost-effectiveness at a £30,000/QALY threshold.

The primary outcome result within the ReDIRECT trial mirrored the HRQoL findings, with self-reported LC symptoms of fatigue, breathlessness and anxiety/depression improving at 6 months in the intervention group compared to the control group[5]. Compared to an age- and sex-adjusted UK general population utility score of 0.86[30], the baseline utility scores in our study population were relatively low, with a mean utility of 0.478 (SD = 0.255) across the two trial arms. For context, a study estimating HRQoL for patients with myalgic encephalomyelitis (ME) and chronic fatigue syndrome (CFS), using the EQ-5D, reported a mean population utility score of 0.469 (95% CI 0.408 – 0.530)[31]. ME/CFS is also considered to be a severely debilitating condition and such low HRQoL scores for this study's LC population provide evidence of the severe impact of LC on those affected.

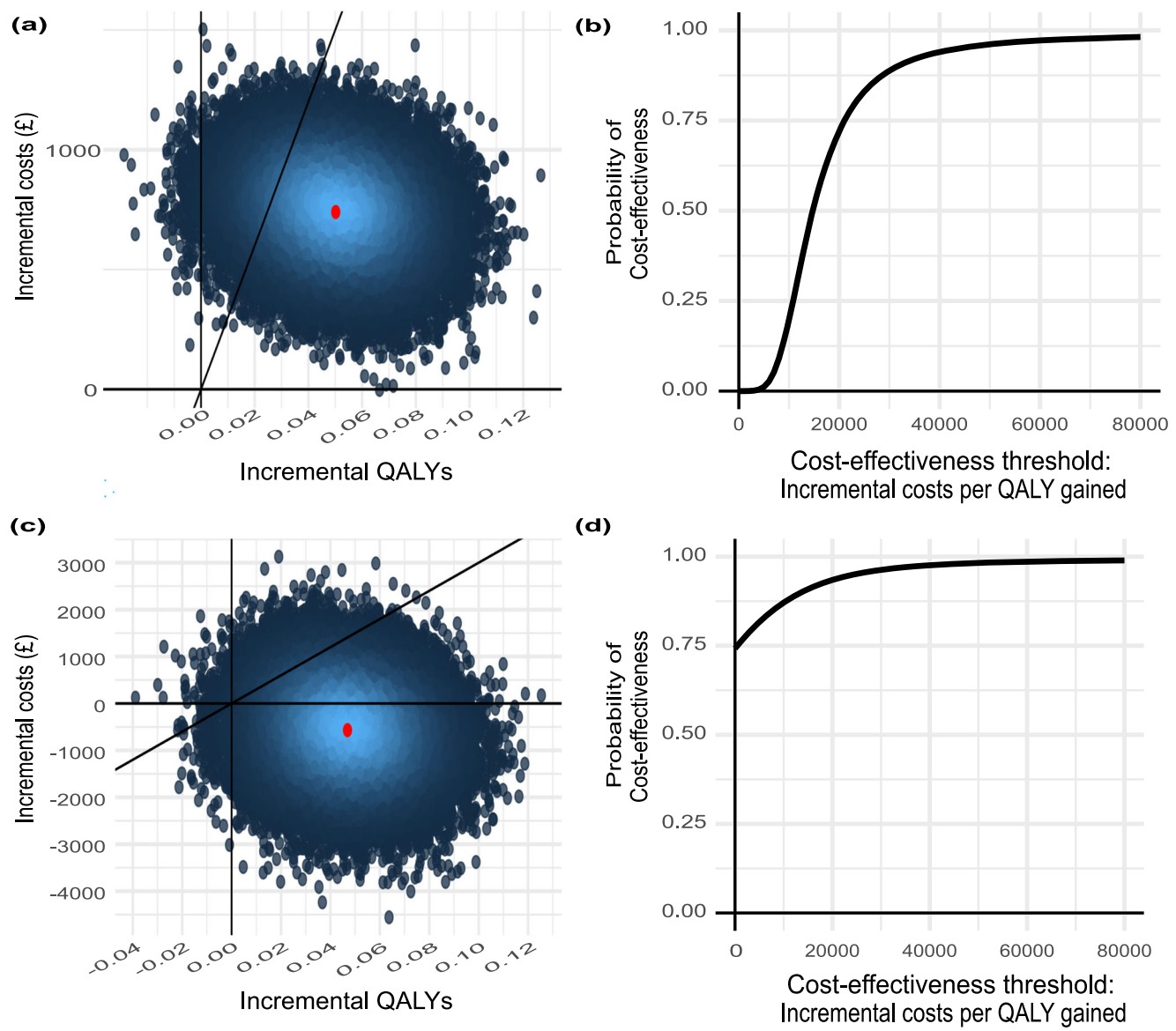

**Fig. 4 | Incremental cost-effectiveness ratio (ICER) Planes and cost-effectiveness acceptability curves (CEACs) for the National Health Service and Personal Social Services (NHS/PSS) and societal perspectives using 5000 non-parametric bootstrap replicates on 10 imputed datasets from an original sample of $n = 234$ individuals ($n = 118$ control, $n = 116$ intervention).** Each individual had three scheduled observations, with a total of 37/702 missing observations (1/354 missing observation in the control arm, 36/348 missing observations in the intervention arm). **a** ICER Plane from the NHS/PSS perspective. **b** CEAC from the NHS/PSS perspective. **c** ICER Plane from the societal perspective. **d** CEAC Plane from the societal perspective. QALY: Quality-adjusted life year.

At 6 months post-randomisation, the incremental QALY gain was 0.0502 (95% CI 0.0127 − 0.0876), with Fig. 1a showing an increase in health utility for the intervention arm at 3 months post-randomisation, and the between-group difference narrowing at 6 months. Other studies evaluating similar remotely delivered weight management interventions report mixed findings relating to HRQoL. A remotely delivered self-directed behavioural intervention (with minimal coaching) among adults with obesity in the US did not impact on HRQoL ($p = 0.81$), with authors suggesting that interventions with behavioural support have greater effect sizes, as the weight loss found in the intervention group was not deemed clinically significant (1.93 kg, 95% CI 0.61 − 3.24)[32]. Another study, conducted in the UK, found that a digital weight loss management intervention showed no evidence of significant weight loss or cost-effectiveness[33]. A lifestyle intervention in the Netherlands involving remote counselling via email found an increase in health utility (measured using the EQ5D) and a 60% probability of being cost-effective at a €20,000/QALY threshold. The

between-group difference in health utility gain (0.01) was not statistically significant (95% CI −0.01 − 0.04), which may have been due to a relatively high baseline mean health utility in this population of 0.913[34]. Important to note is that none of these studies included a total diet replacement component in the intervention, and they did not report clinically meaningful weight loss in the intervention groups when compared to the controls. These comparisons suggest that there may be specific features of the ReDIRECT intervention and study population that play a role in the QALY gain found – particularly the low baseline health utility in our population, the total diet replacement, and the higher level of contact that participants in our study had with dietitians, enhancing the behavioural component of the intervention.

There have been several randomised controlled trials evaluating interventions for the management of LC, with a living systematic review finding evidence of moderate certainty that physical and mental health rehabilitation programmes improve quality of life for people with LC symptoms[35]. However, to the best of our knowledge, no

**Table 4 | Incremental cost-effectiveness results**

| | Perspective | Pooled estimate | 95% CI lower limit | 95% CI upper limit |
|---|---|---|---|---|
| QALY difference | | 0.0501644 | 0.0127108 | 0.0876181 |
| Cost difference (£) | NHS/PSS | 740.11 | 391.97 | 1088.26 |
| | Societal | −568.50 | −2282.34 | 1145.34 |
| ICER | NHS/PSS | £14,754/ QALY | £5,781/QALY | £59,951/ QALY |
| | Societal | Dominant | Dominant | £40,752/ QALY |
| INMB* | NHS/PSS | 765 | −707 | 2237 |
| | Societal | 1976 | −887 | 4839 |

*INMB: incremental net monetary benefit.

other economic evaluation has been published alongside a randomised controlled trial for the management of LC, which makes this study particularly timely and important.

Between-group differences in total healthcare resource use over the trial period were not statistically significant, with an adjusted difference of the intervention group relative to control group of -£264.35 (95% CI: -£640.47 to £ 111.78). While an outlier observation generated higher baseline resource use in the control arm, baseline costs were controlled for in the SUR models, ensuring that these baseline differences do not impact the cost-utility analysis results. Further, sensitivity analysis with this outlier removed did not change interpretation of the cost-utility analysis results.

When including the broader societal perspective costs associated with lost productivity and food costs, the ReDIRECT intervention becomes cost-saving, with an ICER indicating dominance of the intervention over the control. Such a finding indicates the magnitude of the economic burden of LC in terms of absenteeism from work, and demonstrates the potential societal impact of an intervention reducing not only LC symptoms, but also reducing the associated economic burden of LC.

Further to the productivity cost-savings found in the study, the remote delivery mode of the ReDIRECT intervention meant lower intervention costs faced by participants associated with time and travel, compared to interventions delivered at healthcare facilities. The DiRECT trial did not incorporate a societal perspective in the economic evaluation, hence we are unable to draw direct comparisons as to costs faced by participants associated with time, travel and productivity; however, several other studies indicate that remote delivery of interventions result in reduced time, travel and productivity costs faced by patients[36–40]. A nationwide population cohort study for Scotland found an association between socio-economic deprivation and both the risk of developing LC and reduced probability of symptom improvement[41,42]. This indicates the importance of equity considerations for LC and for the implementation of weight management interventions.

Several limitations apply to this within-trial economic evaluation. First, due to mirroring the trial time horizon, there is a relatively short time horizon over which we conducted the economic analysis, with 6 months of data collected for the between-group comparison. It is not yet clear how long symptoms of LC may persist beyond COVID-19 infection, but a decision-analytic model capturing costs and health impacts of weight management over a longer time horizon may provide important economic evidence to inform the implementation of the intervention on a larger scale. The US Panel on Cost-Effectiveness in Health and Medicine recommends that a societal perspective should include future medical costs and effects on future productivity and consumption[43]. Due to the limited time horizon of this within-trial economic evaluation, it was not possible to include such future costs.

With delayed entry into the intervention for the control group at 6 months post randomisation, we only have 6 months of RCT between-group data. However, we have 12 months of observational data for both groups receiving the intervention (the control group was followed for a total of 18 months)[2]. This observational data will be used to validate the extrapolation functions of costs and utilities for the intervention group in the long-term model.

There is some published evidence that health loss can be overstated by the EQ-5D-3L compared to the EQ-5D-5L[44–46]. Therefore, mapping the EQ-5D-5L data collected to the EQ-5D-3L UK value set may have overstated health loss in our population. Future sensitivity analyses with updated UK value sets (when available) using the EQ-5D-5L would be useful to determine the impact of this on our cost-utility analysis results[47].

Using £1000 as the intervention cost for the base-case analysis may not reflect the true delivery cost of the intervention in routine use. The results of the intervention micro-costing analysis indicate that the intervention costs approximately £885 per patient. This micro-costing assumed a conservative cost of £1 per participant for the development of the application, under the assumption that the application had already been developed and implemented by Counterweight Ltd before this study, using established literature for similar interventions with no opportunity costs identified[48]. Further, including application development cost in a within-trial cost-utility analysis with a time horizon of only 6 months means that the cost cannot be annualised, and there is no accounting for potential economies of scale on implementation of the intervention in the NHS, and thus is not appropriate in this analysis. It may be useful to include application development costs as a parameter in a decision analytic model with a longer time horizon and larger study population so that the true impact of this cost input can be reflected.

Unlike the costs, health impacts of adverse events were not explicitly modelled in the within-trial analysis, as it was assumed that any impact would be reflected in the HRQoL of participants, and thus accounted for in utility scores. In a long-term model, it may be important to explicitly account for adverse events by adding an 'adverse event' health state.

In conclusion, this economic evaluation alongside the ReDIRECT clinical trial suggests that weight management may be cost-effective in managing LC symptoms from an NHS/PSS perspective, compared to usual care. When a broader societal perspective is adopted, the intervention is predicted to become cost-saving, dominating usual care, with those in the intervention arm potentially achieving cost savings through productivity gains. In terms of decision-making, this weight management programme meets the health economic criteria for implementation at scale for people living with LC. A decision analytic model estimating cost-effectiveness over the longer term will be useful in informing implementation.

## Methods

### The main randomised controlled trial (ReDIRECT)

ReDIRECT was a wait-list RCT, with participants allocated in a 1:1 ratio at baseline to one of two groups: intervention or control (ISRCTN registry 12595520). Ethical approval of the study protocol was obtained from the South-East Scotland Research Ethics Committee 01 (reference number: 21/SS/0077). Informed consent was obtained from all trial participants.

The allocation was designed to maintain balance with respect to dominant LC symptom (as selected by each participant) and socio-demographic characteristics (sex, ethnicity and index of multiple deprivation). Assessments were taken at baseline, 3 months and 6 months.

Participants in the intervention arm were enroled in the weight management programme, with the intervention described in detail in the trial protocol[2]. In brief, the intervention entailed

dietitian-supported total diet replacement (8–12 weeks), food reintroduction (4–12 weeks), and long-term weight-loss maintenance (up to 1 year). The intervention was delivered remotely by Counterweight Ltd via an online platform with personal video/telephone/in-app chat support. Individuals were allocated a named 'Counterweight Coach' (all registered dietitians) for personal support and regular appointments. All participants could join a coach-moderated in-app 'chat' facility for peer support. Participants allocated to the control arm received usual care and were given access to the weight management programme 6 months after baseline assessment. Usual care involved standard general practitioner (GP) care with potential referral to NHS post-COVID services, typically comprising multidisciplinary assessments, diagnostic tests, and management or appropriate onward referral to post-COVID rehabilitation[49] (the economic evaluation design captured this heterogeneous resource use). Between-group RCT analysis consisted of baseline, three-month and six-month assessments.

The trial primary outcome was a continuous measure derived from the symptom score for the most important LC symptom selected by each participant at baseline (from a list of fatigue, breathlessness, pain, anxiety, depression, and "other"). Secondary outcomes included changes in any of the individual symptoms reported, weight, blood pressure, HRQoL, psychological outcomes, healthcare resource utilisation, work productivity, and personal food costs[2,5]. Process outcomes for the trial will be reported separately.

The study recruited 234 participants aged 18 and above living with overweight/obesity and self-reported symptoms of LC, between December 2021 and July 2022. The inclusion and exclusion criteria for the ReDIRECT Trial were detailed in the published trial protocol[2] with baseline cohort characteristics described in detail elsewhere[15]. The trial was delivered remotely with participants recruited from across the UK (England, Scotland, Wales, and Northern Ireland).

## Economic evaluation
In line with NICE guidance, the base-case within-trial cost-utility analysis of the ReDIRECT intervention compared to usual care, adopted an NHS/PSS perspective[50]. Secondary analysis considered a broader societal perspective (including costs associated with productivity loss, and food expenditure). The time horizon for the within-trial analysis was 6 months, in line with the RCT analysis time horizon. The economic evaluation was carried out according to a pre-specified Health Economics Analysis Plan (HEAP, available upon request), and the updated Consolidated Health Economic Evaluation Reporting Standards (CHEERS) were adhered to in all economic analysis and reporting[51], with a CHEERS checklist available in the Supplementary Materials (Table S1).

## Health outcome for the economic evaluation
The outcome for the economic evaluation was QALYs, estimated using the EQ-5D-5L[52,53], completed at baseline, 3 months, and 6 months to generate AUC QALYs. The scores recorded were mapped to EQ-5D-3L utility values for the UK population using NICE Decision Support Unit guidelines[54]. A cost-utility analysis framework was adopted.

## Resource use
The NHS/PSS perspective base-case analysis included intervention costs and direct health and personal social service use by participants, while the broader societal perspective also captured indirect costs associated with productivity (employment) impacts, and personal food costs. All costs were estimated in 2022 British pound sterling.

**Intervention costs.** In the primary analysis, the intervention cost was set at the cost to the NHS of utilising the Counterweight-Plus intervention (£1000). Micro-costing of resource use associated with the

delivery of the intervention was conducted as a scenario analysis, with the unit cost inputs for each resource item presented in Table S2 in the Supplementary Materials.

**Healthcare service resource use.** Data on healthcare services used by participants in the preceding 3 months were collected at baseline then at 3 months and 6 months following randomisation. These data include consultations with general practitioners (GPs), nurses, allied healthcare professionals, community nurses and social workers. Data on hospital visits (both outpatient visits and in-patient stays) were also collected. The quantity of each resource use item was combined with their unit cost to estimate total healthcare resource use per participant and mean healthcare resource use for each trial arm. Unit costs attached to healthcare resource use were obtained from various sources including the Personal Social Services Research Unit (PSSRU), NHS reference costs, Scottish Health Service costs, the Office for National Statistics, and the literature (see Table 5)[55–62]. Data pertaining to medication use were collected for each participant at baseline, three and 6 months. Unit costs from the British National Formulary (BNF) were used to estimate the cost of medication use for each trial arm over the trial period[63,64].

**Adverse events.** All adverse events throughout the trial were recorded[2]. Healthcare resource use reported by participants was cross-checked with all adverse events logged to ensure all resource use associated with potential adverse events was measured and valued for inclusion in the cost-utility analysis.

**Productivity costs and personal food costs.** Data were collected on hours of work missed due to illness in the preceding seven-day period at baseline, three and 6 months using the Work and Productivity Activity Impairment (WPAI) instrument[65]. Costs associated with lost productivity (indirect costs) were estimated for each arm over the trial period: age- and sex-specific national average weekly wage rates applied to the mean hours missed per week and extrapolated over the three-month period preceding each assessment. Similarly, participants were asked about their household food costs in the preceding seven-day period at baseline, 3 months and 6 months, as well as the number of individuals in the household. This was used to estimate a per-person food cost by trial arm, with mean weekly costs extrapolated over the three-month period preceding each assessment during the study period. The WPAI instrument and personal food and drink costs questionnaire can be found in the Supplementary Materials.

## Missing data
Missing data were descriptively analysed, and MICE was conducted, separating the intervention and control arm[28,66]. Missing cost data were imputed at the level of the main resource use categories. Missing EQ-5D-5L data were imputed at the level of the utility score[66,67]. SUR models were fitted on the imputed datasets and for each imputed dataset 5000 bootstrapped samples were drawn[68,69]. The SUR modes included age, sex, index of multiple deprivation, baseline costs, baseline utilities and the primary outcome selected, in line with the statistical analysis conducted for the trial primary outcome. The cost and health outcome statistics of interest were then pooled using Rubin's rules, and pooled covariance estimated[66,68]. An ICER plane and CEAC were derived from the output[70].

## Incremental cost-utility analysis
The results of the cost-utility analysis are presented as incremental costs per incremental QALY gained. An ICER was calculated as the between-group difference in the total costs divided by the difference in QALYs at 6 months post-randomisation. 95% confidence intervals were based on 5000 non-parametric bootstrap iterations. An ICER plane and CEAC were drawn from the results of the multiple

**Table 5 | Unit costs used for estimation of healthcare resource use costs, intervention micro-costing and indirect costs**

| Item | Unit cost (£) | Reference/source |
|---|---|---|
| **Healthcare service resource use** | | |
| GP visit | 42 | Personal Social Services Research Unit[57] |
| GP telephone call | 41.13 | Personal Social Services Research Unit[57] |
| Nurse visit | 11.31 | Literature[56] |
| Nurse telephone call | 11.31 | Literature[56] |
| Specialist LC nurse | 28.5 | Personal Social Services Research Unit[57] |
| Physiotherapist | 73.14 | NHS Reference costs[58] |
| Occupational therapist | 99.11 | NHS Reference costs[58] |
| Community nurse | 23 | Personal Social Services Research Unit[57] |
| Health visitor | 94.25 | NHS reference costs[58] |
| Social worker | 50 | Personal Social Services Research Unit[57] |
| Home care | 23 | Personal Social Services Research Unit[57] |
| Hospitalisation (overnight) | 586.59 | Literature[59] |
| A&E visit | 148 | Scottish Health Services Costs[60] |
| Day hospital visit | 251 | Scottish Health Services Costs[60] |
| Outpatient department appointment | 165 | NHS Reference costs[58] |
| **Societal perspective** | | |
| National average weekly wage | Age- and sex-specific | Office for National Statistics[61,62] |

imputations and presented for both the base case NHS/PSS perspective as well as the broader societal perspective.

## Software

For data collection, trial participants entered data directly into bespoke electronic Case Report Forms (eCRF) and data relating to dietetic visits were entered in Excel spreadsheets. All analyses were conducted using R Statistical Software, version 4.4.1 (2024-06-14 ucrt) and RStudio version 2024.04.2−764[71,72]. The R packages used in the analysis include: boot (R package version 1.3−30)[73], data.table (R package version 1.16.0)[74], lme4 (R package version 1.1-35.5)[75], mice (R package version 3.16.0)[28], summarytools (R package version 1.0.1)[76], ggpattern (R package version 1.1.4)[77], readstata13 (R package version 0.10.1)[78], viridis (package version 0.6.5)[79], flextable (R package version 0.9.6)[80], finalfit (R package version 1.0.8)[81], systemfit (R package version 1.1-30)[69], doBy (R package version 4.6.22)[82], ggpubr (R package version 0.6.0)[83], ggpointdensity (R package version 0.1.0)[84], eq5d (R package version 0.15.3)[85], table1 (R package version 1.4.3)[86], miceadds (R package version 3.17)[87], gtsummary (R package version 2.0.2)[88], reshape (R package version 0.8.9)[89], tidyverse (R package version 2.0.0)[90], devtools (R package version 2.4.5)[91], scales (R package version 1.3.0)[92], rmarkdown (R package version 2.28)[93], kableExtra (R package version 1.4.0)[94].

## Reporting summary

Further information on research design is available in the Nature Portfolio Reporting Summary linked to this article.

## Data availability

As per our study protocol, access to the raw data is restricted to the primary research team while the research is being conducted and prior to publication of the primary research papers. The primary research papers include the primary trial outcome paper (published, ref. 5), this health economics paper, the process evaluation paper and the Patient and Public Involvement paper. Upon publication of these papers, fully anonymized and minimised data (and data dictionaries) will be placed in a research data repository with access given to bona fide researchers, on request to the corresponding author and subject to appropriate data-sharing agreements. Proposals will be assessed on a monthly basis, with a response within 2 months of submission. The

Trial Statistical Analysis Plan is available via ISRCTN (10.1186/ISRCTN12595520), and ReDIRECT study eCRF questionnaire screenshots are available via Figshare at 10.6084/m9.figshare.21270837 (ref. 95). Source data for Figs. 1–3 and Fig. S1 are provided with this paper.

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

## Acknowledgements

The authors thank the National Health Service (NHS Greater Glasgow and Clyde) sponsor representatives, project management unit staff, Robertson Centre for Biostatistics and ReDIRECT Trial Steering Committee members for their support. The authors are also grateful to Long COVID Scotland and patient representatives in the PPI group for constructive discussions about the protocol, data acquisition and interpretation. The authors thank all ReDIRECT study participants for their participation in research. This study (COV-LT2-0059 DB) was funded by the National Institute for Health and Care Research (NIHR) in response to the COVID-19 pandemic. The views expressed are those of the authors and not necessarily those of the NIHR or the Department of Health and Social Care. The study's funder had no role in study design, data collection, analysis, interpretation or report writing. The corresponding authors had final responsibility for the decision to submit for publication.

## Author contributions

E.C. and D.N.B. are the principal investigators of the study, with E.M. the principal health economist of the study. The study was conceptualised by E.C., D.N.B., M.L., A.M., E.M., C.A.O., J.O., N.S. and T.I. All authors contributed to the methodology. L.H. and J.R. coordinated the recruitment and quantitative data acquisition, supervised by E.C. and D.N.B. Processing and quality assurance of quantitative data was conducted by C.H., supervised by A.M., and N.B. coordinated Counterweight data acquisition. H.L.F. carried out formal analysis and data visualisation, supervised by E.M. H.L.F. wrote the original draft, which was reviewed by all authors. All authors had full access to all draft versions and supplementary materials and had final responsibility for the decision to submit for publication.

## Competing interests

N.B. is an employee and shareholder of Counterweight Ltd., sub-contracted to the University of Glasgow to deliver the ReDIRECT intervention. A.M. is a member of Clinical Steering Committee for ARC Medical Inc. N.S. has received institutional grant support from Astra-Zeneca, Boehringer Ingelheim, Novartis, Roche Diagnostics and honoraria from Abbott Laboratories, AbbVie, Afimmune, Amgen, Astra-Zeneca, Boehringer Ingelheim, Eli Lilly, Hanmi Pharmaceuticals, Janssen, Menarini-Ricerche, Merck Sharp & Dohme, Novartis, Novo Nordisk, Pfizer, Roche Diagnostics, Sanofi. M.L. has received lecturing fees from Novo Nordisk, Lilly, Nestle, Oviva, Merck, and Sanofi and is a medical advisor to Counterweight Ltd, with fees paid to the University of Glasgow. The remaining authors declare no competing interests.
