## [Transparent Peer Review file · Nature Communications]

Cost-Effectiveness of the ReDIRECT/Counterweight-Plus Weight Management Programme to Alleviate Symptoms of Long COVID

Corresponding Author: Ms Heather Fraser

Version 0:

Reviewer comments:

Reviewer #1

(Remarks to the Author)

Thank you for the opportunity to review this study. I have enjoyed reading it and I hope the feedback is helpful for improving the manuscript.

1. [Table 1] Authors should report the baseline healthcare cost and productivity loss for both groups in Table 1, particularly given that there are substantial differences between groups for these outcomes.
2. ['Healthcare resource use' section under Results] Can you clarify whether the healthcare resource use outliers in the control group accounted for the significant difference in healthcare costs between the two groups at baseline and 3-month follow-up?
3. ['Healthcare resource use' section under Results] What is the mean difference in healthcare cost between groups? Only the 95% confidence interval is given.
4. [Results] Authors should provide a separate sub-heading for the section commencing "Figure 4(a) and 4(c)..." Perhaps "Cost-effectiveness analysis"?
5. [Discussion] The third paragraph summarising the results of previous studies of weight loss and lifestyle interventions are very helpful. I wonder if there are any previous randomised control trials that targeted similar LC populations (lifestyle or pharmacological interventions)? It'd be helpful to put the current study's results in that context. If there are no previous trials, this should be emphasised as a key strength of this study.
6. [Discussion] "This illustrates the potential of remotely delivered weight management to improve equity in the distribution of health benefits" and similar statement in last paragraph of Discussion. We wouldn't know this without data on the update of the intervention disaggregated by socioeconomic status subgroups. It's possible that wealthier and more educated persons with higher digital literacy take up the intervention more frequently. I suggest you remove these statements or support them with further data.

Reviewer #2

(Remarks to the Author)

The manuscript is a well-written report on the cost-effectiveness analysis (CEA) of the ReDIRECT trial; results of the primary outcomes of the trial have been accepted for publication in Nature Medicine. My main comment is to remove the ICERs for the cost-saving results.

A CEA combines the results of two variables: 1) difference in cost between the intervention and control groups, and 2) difference in outcomes between the intervention and control groups. The combination can occur in one of four quadrants. An incremental cost-effectiveness ratio (ICER) is generally reported for an increase in costs and an improvement in outcomes (i.e. the northeast quadrant). The ratio of a decrease in cost and improvement in outcomes (i.e. the southeast quadrant) can't be interpreted. Please see for an explanation in Stinnett AA, Mullahy J. The negative side of cost-effectiveness analysis. JAMA 1997; 277: 1931–2; author reply 1931-1932. The authors can simply report that the result is "dominant" or "cost-saving" and percentage of the results in Figure 4c that are in the south-east quadrant. Specifically:

Line 40. Delete the phrase, “with an ICER of -£11,319/QALY”

Table 4. In the row with ICER for societal perspective, delete “-£11,319/QALY”, and replace “£81,357/QALY” with “dominant”

Line 215. Delete the phrase in parentheses.

Line 257. Replace “even more cost-effective” with “cost-saving.”

Other comments are:

1. I invite the authors to consider adding two references on LC to the introduction:

A. Global estimates of the proportion of individuals with LC symptoms, Global Burden of Disease Long COVID Collaborators. Estimated global proportions of individuals with persistent fatigue, cognitive, and respiratory symptom clusters following symptomatic COVID-19 in 2020 and 2021. JAMA. 10 October 2022. doi: 10.1001/jama.2022.18931.

B. Recent synthesis of evidence on LC, Al-Aly Z, Davis H, McCorkell L, et al. Long COVID science, research and policy. Nature Medicine. 9 August 2024. doi: 10.1038/s41591-024-03173-6.

2. Table 1. Report descriptive statistics on all the covariates in the regression analyses, including index of multiple deprivation, ethnicity, participant-selected primary outcome at baseline, hours of work missed due to sickness at baseline, all resource use at baseline.

3. Line 100. Briefly explain that £1000 is the fee that Counterweight-Plus charges the NHS, and therefore more appropriate for the NHS/PSS perspective than the £885 result of the micro-costing analysis.

4. Lines 113 and 207. Report the mean difference in healthcare resource use in addition to the 95% CI.

5. Table 4. Define the term INMB. Is it incremental net monetary benefit?

6. Line 207. Discuss the adjusted total results in Table 3 rather than repeating the six-month result.

7. Line 85. There is evidence that the EQ-5D-3L over states health loss relative to the EQ-5Q-5L. (Thompson AJ and Turner AJ. Pharmacoeconomics 2020, Marti-Pastor et al. Population Health Metrics 2018, Janssen et al. Pharmacoeconomics 2018. Although the authors collected data using the EQ-5D-5L, discuss how mapping those data to the EQ-5D-3L UK value set would potentially overstate health loss in their analysis.

8. Lines 219-221. I would welcome the authors estimates on the cost of the remote delivery mode of the ReDIRECT intervention compared to the DIRECT intervention.

9. Line 227. In the limitations, note that the authors' measure of social cost is not universally accepted. The US Panel on Cost-Effectiveness in Health and Medicine recommended a broader definition that includes future medical costs, and future consumption unrelated to health. See Sanders JD et al. JAMA 2016.

Reviewer #3

(Remarks to the Author)

Overview

This is a concisely written, well executed and reproducible within-trial economic evaluation of a weight management programme for the treatment of long COVID. It is good to see an economic evaluation conducted in a UK context that considers productivity/broader societal impacts. I think this inclusion makes the evaluation particularly noteworthy given the described societal impacts of long COVID. Showing that this is a good use of healthcare resources will be of interest to readers. I also like the broader links to myalgic encephalomyelitis and chronic fatigue syndrome; I think they place this analysis in a broader context. On the whole, it is difficult to find much to critique. I have however noted a few questions/points.

Questions

- I would be interested to hear why the mean resource use and mean hours of work missed for the intervention and comparator are so different at baseline. I would expect them to be the same at baseline or perhaps mean resource use to be higher as there is the cost of treatment to consider. A bit of context on what is driving the difference would be helpful. I appreciate the authors have adjusted for this in statistical analysis.

- Regarding the inclusion of productivity and food costs into the economic evaluation, I appreciate the consideration of these potentially important outcomes/costs. I am unsure whether their inclusion in a net benefit framework is methodologically sound, although I am sure this is what is done in the literature. If I have understood the methods correctly, productivity and food costs are included on the cost side of the calculation meaning net benefit assumes the opportunity costs of productivity and food consumption are assumed to fall on the health care budget. This may be unimportant for the results as they are both cost-saving but I would be interested to hear from the authors regarding whether they think this is methodologically justified.

- Is there a reason why the methods section starts on Line 264, Page 11?

Version 1:

Reviewer comments:

Reviewer #1

(Remarks to the Author)

I am satisfied with the clarifications and amendments in response to the comments.

Reviewer #2

(Remarks to the Author)

I am grateful for the author's response to my initial comments and revisions to the manuscript, and have no further comments. Congratulations on a very good analysis and well-written manuscript.

Reviewer #3

(Remarks to the Author)

I would like to thank the authors for their responses. Overall, my concerns have been addressed and have no further concerns.

March 21, 2025

Dear Editors and Reviewers,

Thank you for your comments and suggestions. We feel that they have significantly improved our manuscript.

We have included a table below, outlining our responses to each comment. We have also attached a revised manuscript and supplementary materials with tracked changes, so that edits can be clearly identified.

Sincerely,
The authors

Response to reviewers' comments

REVIEWER COMMENTS

Reviewer #1 (Remarks to the Author):

Thank you for the opportunity to review this study. I have enjoyed reading it and I hope the feedback is helpful for improving the manuscript.

Section	Comment	Response
Table 1 (Results)	Authors should report the baseline healthcare cost and productivity loss for both groups in Table 1, particularly given that there are substantial differences between groups for these outcomes.	Thank you for this suggestion, healthcare resource use costs and productivity loss at baseline for both groups have been added into Table 1.
'Healthcare resource use' section under Results	Can you clarify whether the healthcare resource use outliers in the control group accounted for the significant difference in healthcare costs between the two groups at baseline and 3-month follow-up?	Thank you for this question. To clarify, the healthcare resource use outliers in the control group accounted for a large amount of the difference in healthcare costs at baseline, but did not account for the difference in healthcare costs between the groups at 3-month follow up. This gives further confidence that controlling for resource use at baseline ensures that these baseline differences do not impact the cost-utility analysis results. Further detail on this point has been added to the outlier sensitivity analysis section in the Supplementary Materials, where the healthcare resource use at each time point, by trial arm is shown in Figure S1. Further a table (Table S3) has been added, allowing a side-by side comparison

		of the mean healthcare costs at each visit, by trial arm, for the base case analysis compared to the scenario analysis with the outlier removed.
'Healthcare resource use' section under Results	What is the mean difference in healthcare cost between groups? Only the 95% confidence interval is given.	Thank you for this question. We have now added in the mean difference in healthcare costs between groups, with the added text (lines 122-124) reading: "Differences in healthcare resource use in the intervention arm relative to the control arm were not statistically significant at six months (mean difference: £15.79, 95% CI: -£147.30 to £178.88). Previously, we had mis-stated the timeline, referring to reporting total healthcare resource use difference over the '6 month trial period', not at six months. Please note we have now corrected this to reporting the difference at six months, with the discussion section (lines 227-229) reporting the difference over the six-month trial period.
Results	Authors should provide a separate sub-heading for the section commencing "Figure 4(a) and 4(c)..." Perhaps "Cost-effectiveness analysis"?	Thank you for this suggestion, we have added in the sub-heading as suggested.
Discussion	The third paragraph summarising the results of previous studies of weight loss and lifestyle interventions are very helpful. I wonder if there are any previous randomised control trials that targeted similar LC populations (lifestyle or pharmacological interventions)? It'd be helpful to put the current study's results in that context. If there are no previous trials, this should be emphasised as a key strength of this study.	Thank you for the helpful comment and we completely agree this would have provided excellent context for our results. While there have been several RCTs for LC, particularly targeting physical rehabilitation, we have not been able to find any similar economic evaluations – we have added text (lines 220-225) to include both of these points:

		“There have been several randomised controlled trials evaluating interventions for the management of LC, with a living systematic review finding evidence of moderate certainty that physical and mental health rehabilitation programmes improve quality of life for people with LC symptoms (34). However, to the best of our knowledge, no other economic evaluation has been published alongside a randomised controlled trial for the management of LC, which makes this study particularly timely and important.”
Discussion	“This illustrates the potential of remotely delivered weight management to improve equity in the distribution of health benefits” and similar statement in last paragraph of Discussion. We wouldn’t know this without data on the uptake of the intervention disaggregated by socioeconomic status subgroups. It’s possible that wealthier and more educated persons with higher digital literacy take up the intervention more frequently. I suggest you remove these statements or support them with further data.	Thank you for pointing this out. We have removed these two statements.

Reviewer #2 (Remarks to the Author):

The manuscript is a well-written report on the cost-effectiveness analysis (CEA) of the ReDIRECT trial; results of the primary outcomes of the trial have been accepted for publication in Nature Medicine. My main comment is to remove the ICERs for the cost-saving results.

Section	Comment	Response
	A CEA combines the results of two variables: 1) difference in cost between the intervention and control groups, and 2) difference in outcomes between the intervention and control groups. The combination can occur in one of four quadrants. An incremental cost-effectiveness ratio (ICER) is generally reported for an increase in costs and an improvement in outcomes (i.e. the northeast quadrant). The ratio of a decrease in cost and improvement in outcomes (i.e. the southeast quadrant) can't be interpreted. Please see for an explanation in Stinnett AA, Mullahy J. The negative side of cost-effectiveness analysis. JAMA 1997; 277: 1931–2; author reply 1931-1932. The authors can simply report that the result is “dominant” or “cost-saving” and percentage of the results in Figure 4c that are in the south-east quadrant. Specifically:	Thank you for sending this helpful explanation, I agree it makes more sense to remove the negative ICERs as they cannot be interpreted and have done so as suggested through the text.
Abstract	Line 40. Delete the phrase, “with an ICER of -£11,319/QALY”	Removed, thank you.
Results	Table 4. In the row with ICER for societal perspective, delete “-£11,319/QALY”, and replace “£81,357/QALY” with “dominant”	Done, thank you.
Discussion	Line 215. Delete the phrase in parentheses.	Removed, thank you.

Discussion	Line 257. Replace “even more cost-effective” with “cost-saving.”	Done, thank you.
Introduction	1. I invite the authors to consider adding two references on LC to the introduction: A. Global estimates of the proportion of individuals with LC symptoms, Global Burden of Disease Long COVID Collaborators. Estimated global proportions of individuals with persistent fatigue, cognitive, and respiratory symptom clusters following symptomatic COVID-19 in 2020 and 2021. JAMA. 10 October 2022. doi: 10.1001/jama.2022.18931. B. Recent synthesis of evidence on LC, Al-Aly Z, Davis H, McCorkell L, et al. Long COVID science, research and policy. Nature Medicine. 9 August 2024. doi: 10.1038/s41591-024-03173-6.	Thank you for these suggestions, the following texts have been added to the introduction (lines 51-54 and lines 60-61), and specific citations included: “A Bayesian meta-regression of 54 studies and two databases estimated that, of individuals surviving symptomatic episodes of COVID-19 infection, 6.2% experienced at least one LC symptom three months after initial infection, with 15.1% of these individuals experiencing persistent symptoms at 12 months⁽⁶⁾.” “LC has been found to affect labour participation, employment and productivity of individuals as well as their caregivers ⁽¹⁴⁾.”
Results	Table 1. Report descriptive statistics on all the covariates in the regression analyses, including index of multiple deprivation, ethnicity, participant-selected primary outcome at baseline, hours of work missed due to sickness at baseline, all resource use at baseline.	Thank you for the suggestion, all covariates in the regression analyses are now reported in Table 1.
Results	Line 100. Briefly explain that £1000 is the fee that Counterweight-Plus charges the NHS, and therefore more appropriate for the NHS/PSS perspective than the £885 result of the micro-costing analysis.	Thank you for the suggestion, we have added that in, the first sentence of the “Intervention micro-costing” section (lines 106-108) now reads: “Counterweight Ltd charges the NHS £1,000 per person for delivery of the intervention, which was therefore deemed the appropriate intervention cost to use in the base case analysis (rather than the micro-cost estimate).”

Results and Discussion	Lines 113 and 207. Report the mean difference in healthcare resource use in addition to the 95% CI.	Thank you for the suggestion, those lines (122-124 and 226-229) now read: “Differences in healthcare resource use in the intervention arm relative to the control arm were not statistically significant at six months (mean difference: £15.79, 95% CI: -£147.30 to £178.88).” And: “Between-group differences in total healthcare resource use over the trial period were not statistically significant, with an adjusted difference of the intervention group relative to control group of -£174.38 (95% CI: -£492.37 to £143.61).” Previously, we had mis-stated the timeline, referring to reporting total healthcare resource use difference over the ‘6 month trial period’, not at six months. Please note we have now corrected this to reporting the difference at six months, with the discussion section (lines 226-229) reporting the difference over the six-month trial period.
Results	Table 4. Define the term INMB. Is it incremental net monetary benefit?	Thank you for pointing this out, the definition (Incremental net monetary benefit) has now been added to the foot of Table 4.
Discussion	Line 207. Discuss the adjusted total results in Table 3 rather than repeating the six-month result.	Thank you for pointing out the repetition, this has been modified as suggested. Previously, we had mis-stated the timeline, referring to reporting total healthcare resource use difference over the ‘6 month trial period’, not at six months. Please note we have now corrected this to reporting the difference at six months, with the

		discussion section (lines 226-229) reporting the difference over the six-month trial period.
Results	Line 85. There is evidence that the EQ-5D-3L over states health loss relative to the EQ-5Q-5L. (Thompson AJ and Turner AJ. Pharmacoeconomics 2020, Marti-Pastor et al. Population Health Metrics 2018, Janssen et al. Pharmaceconomics 2018. Although the authors collected data using the EQ-5D-5L, discuss how mapping those data to the EQ-5D-3L UK value set would potentially overstate health loss in their analysis.	Thank you for this comment, this is an important observation. We have added the following to the 'limitations' section in the discussion (lines 265-269): “There is some published evidence that health loss can be over-stated by the EQ-5D-3L compared to the EQ-5D-5L (43-45). Therefore, mapping the EQ-5D-5L data collected to the EQ-5D-3L UK value set may have overstated health loss in our population. Future sensitivity analyses with updated UK value sets (when available) using the EQ-5D-5L would be useful to determine the impact of this on our cost-utility analysis results (46).”
Discussion	Lines 219-221. I would welcome the authors estimates on the cost of the remote delivery mode of the ReDIRECT intervention compared to the DIRECT intervention.	The DiRECT economic evaluation did not incorporate a societal perspective, and thus did not estimate costs faced by participants associated with time, travel and productivity, and so we are unable to draw a direct comparison. We have clarified this in the text and added the following (lines 242-246): “The DiRECT trial did not incorporate a societal perspective in the economic evaluation, hence we are unable to draw direct comparisons as to costs faced by participants associated with time, travel and productivity; however, several other studies indicate that remote delivery of interventions result

		in reduced time, travel and productivity costs faced by patients ⁽³⁴⁻³⁸⁾ .”
Discussion	Line 227. In the limitations, note that the authors’ measure of social cost is not universally accepted. The US Panel on Cost-Effectiveness in Health and Medicine recommended a broader definition that includes future medical costs, and future consumption unrelated to health. See Sanders JD et al. JAMA 2016.	Thank you for this suggestion. The following text has been added to the limitations section in the discussion (lines 256-259): “The US Panel on Cost-Effectiveness in Health and Medicine recommends that a societal perspective should include future medical costs and effects on future productivity and consumption⁽⁴²⁾. Due to the limited time horizon of this within-trial economic evaluation, it was not possible to include such future costs.”

Reviewer #3 (Remarks to the Author):

Overview

This is a concisely written, well executed and reproducible within-trial economic evaluation of a weight management programme for the treatment of long COVID. It is good to see an economic evaluation conducted in a UK context that considers productivity/broader societal impacts. I think this inclusion makes the evaluation particularly noteworthy given the described societal impacts of long COVID. Showing that this is a good use of healthcare resources will be of interest to readers. I also like the broader links to myalgic encephalomyelitis and chronic fatigue syndrome; I think they place this analysis in a broader context. On the whole, it is difficult to find much to critique. I have however noted a few questions/points.

Section	Comment	Response
	I would be interested to hear why the mean resource use and mean hours of work missed for the intervention and comparator are so different at baseline. I would expect them to be the same at baseline or perhaps mean resource use to be higher as there is the cost of treatment to consider. A bit of context on what is driving the difference would be helpful. I appreciate the authors have adjusted for this in statistical analysis.	Thank you for your kind comments and interesting questions. The mean resource use is higher at baseline in the control arm largely because of one outlier who had a lengthy hospital stay, whereas there were very few hospital visits in the intervention arm. It was on this basis that we conducted the outlier sensitivity analysis, to get an understanding of the impact of this individual on the economic analysis results and interpretation. The intervention costs were presented separately to the healthcare resource use costs, but when they are included in the analysis, total costs (from an NHS/PSS perspective) in the intervention arm are higher than the control arm. With work and productivity, it is more challenging to define exactly why these costs are higher in the control arm compared to the intervention arm. We were unable to

		identify an outlier, or any particular reason for these baseline differences. For example, baseline employment status is similar between groups (see Table 1). As you rightly point out, we have adjusted the statistical analysis to control for these costs at baseline to reduce the impact on the overall results.
	Regarding the inclusion of productivity and food costs into the economic evaluation, I appreciate the consideration of these potentially important outcomes/costs. I am unsure whether their inclusion in a net benefit framework is methodologically sound, although I am sure this is what is done in the literature. If I have understood the methods correctly, productivity and food costs are included on the cost side of the calculation meaning net benefit assumes the opportunity costs of productivity and food consumption are assumed to fall on the health care budget. This may be unimportant for the results as they are both cost-saving but I would be interested to hear from the authors regarding whether they think this is methodologically justified.	Thank you for this question. For the societal perspective only, we included productivity and food costs on the cost side of the calculation. We agree with your point that, from an NHS/PSS perspective (that of the health budget in the UK), it would not be methodologically justified to include work and productivity and food costs, which fall to the individual.
	Is there a reason why the methods section starts on Line 264, Page 11?	Yes, apologies for the confusion, the Nature Communications submission guidelines ask authors to submit manuscripts with a section order of: Introduction, Results, Discussion and then Methods.